# Zn and Zn-Fe Nanostructures with Multifunctional Properties as Components for Food Packaging Materials

**DOI:** 10.3390/nano12122104

**Published:** 2022-06-18

**Authors:** Hafsae Lamsaf, Lina F. Ballesteros, Miguel A. Cerqueira, José A. Teixeira, Lorenzo M. Pastrana, Luís Rebouta, Sandra Carvalho, Sebastian Calderon

**Affiliations:** 1CF-UM-UP, Centre of Physics of Minho and Porto Universities, Campus of Azurém, 4800-058 Guimarães, Portugal; hafsae.lamsaf@gmail.com (H.L.); rebouta@fisica.uminho.pt (L.R.); 2INL—International Iberian Nanotechnology Laboratory, Av. Mestre José Veiga s/n, 4715-330 Braga, Portugal; miguel.cerqueira@inl.int (M.A.C.); lorenzo.pastrana@inl.int (L.M.P.); 3CEB—Centre of Biological Engineering, University of Minho, Campus of Gualtar, 4710-057 Braga, Portugal; linafernanda37@ceb.uminho.pt (L.F.B.); jateixeira@deb.uminho.pt (J.A.T.); 4LABBELS–Associate Laboratory, Braga/Guimarães, Portugal; 5CEMMPRE, Mechanical Engineering Department, University of Coimbra, 3030-788 Coimbra, Portugal; sandra.carvalho@dem.uc.pt

**Keywords:** zinc–iron, nanostructure, sputtering, food packaging, chromatic effect, antibacterial activity

## Abstract

Metallic and bimetallic nanostructures have shown interesting chromatic and antibacterial properties, and they can be used in various applications. In this work, zinc (Zn) and iron (Fe) nanostructures were produced with different morphologies: (i) pure Zn; (ii) Zn-Fe nanoalloys; (iii) Zn-Fe nanolayers (Zn-Fe NLs); and (iv) Zn nanolayers combined with Fe nanoparticles (Zn NLs + Fe NPs). The aim was to produce components for food packaging materials with active and intelligent properties, including oxygen absorption capacity, chromatic properties, and antibacterial properties. Thus, the morphology, structure, and chemical composition of the samples were characterized and correlated with their oxidation, chromatic, and antibacterial properties. The results revealed a relevant reduction in the coating’s opacity after oxidation varying from 100 to 10% depending on the morphology of the system. All coatings exhibited significant antibacterial activity against *S. aureus*, revealing a direct correlation with Zn content. The incorporation of Fe for all atomic arrangements showed a negative impact on the antibacterial effect against *E. coli*, decreasing to less than half the zone of inhibition for Zn-Fe NLs and Zn NLs + Fe NPs and suppressing the antibacterial effect for Zn-Fe alloy when compared with the pure Zn system.

## 1. Introduction

The preservation of food quality and the extension of its shelf life have become the main subject of many investigations in the field of packaging materials. Consequently, food packaging manufacturers have developed multifunctional materials that can guarantee food quality, shelf-life extension, cost efficiency, product safety, and consumer demand. Thus, to achieve such multifunctional requirements, researchers and industries have focused on the development of novel nanomaterials compatible with food products. Some examples of these are metallic nanoparticles, which have been explored as antibacterial agents [1,2,3], antioxidants [4,5,6], catalyzers [7,8], and photocatalytic and scavenging mediators [3].

Nanoparticles (NPs) of transition metals have demonstrated potential in technological applications compared with macroscopic materials, particularly when used in food packaging, enhancing the mechanical properties and controlling the biodegradability of the produced materials [3,9,10]. Moreover, studies have confirmed the benefit that such NPs bring to consumers, since the NPs contribute to extending the shelf life of food, ensuring better traceability and providing reinforced protection [11,12]. Several metals, such as zinc (Zn), titanium (Ti), iron (Fe), and copper (Cu), have been used to produce NPs for food packaging materials [13,14], acting as antimicrobial agents, pigments, and oxygen scavengers [1,15,16]. 

Zinc oxide (ZnO) NPs, known for their multifunctional effects, are currently being studied in food packaging materials as antibacterial agents, for preventing food contamination due to harmful bacteria [1], and as absorbers of ultraviolet light (UV), taking advantage of the wide bandgap (Eg = 3.37 eV) of ZnO [17]. Shankar et al. [18], for instance, showed an increase in minced fish cake shelf life and strong antibacterial activity against foodborne pathogenic bacteria, *Escherichia coli*, and *Listeria monocytogenes* for poly (lactic acid) and ZnO (PLA/ZnO) NP composite films. Moreover, in previous research by the authors of the current study, Zn NPs in the metallic state demonstrated oxygen scavenging properties at high humidity, useful for food packaging applications [19]. Fe, on the other hand, is a known colorant in food [20] and has been used in nanopowder form as an oxygen scavenger in low- and high-relative humidity environments, with a scavenging rate three times higher than that of microscavengers when exposed at 100% relative humidity [21]. 

Bimetallic nanostructures have been shown to enhance the functionality of metallic NPs, providing not only the individual properties of the components but synergetic new phenomenology due to physical binding between the metals. Consequently, combining Zn and Fe metallic nanostructures was expected to provide multifunctional characteristics to food packaging materials. Fe-Zn oxide, for example, has demonstrated good magnetic and antibacterial properties, depending on its composition and morphology. Gordon et al. [22] showed that higher ratios of Zn/Fe NPs had more important antibacterial activity against *Staphylococcus aureus* and *Escherichia coli*. Furthermore, recent work on galvanic oxidation of bimetallic Zn-Fe NPs for oxygen scavenging [23] revealed that the bimetallic system of Zn-Fe accelerated the oxidation mechanism. The obtained material presented great potential to be used as an oxygen scavenger [23], as precise control of the morphology could be obtained by using magnetron sputtering techniques [24].

The morphology of NPs is another factor that strongly influences their multifunctional properties, and therefore, a convenient production method for NPs needs to be chosen. NPs are usually prepared using physical, chemical, and biological methods [25,26,27], which can influence their geometry, aspect ratio, and distribution [3,28,29] as well as their composition and toxicity. However, the incorporation of the nanoparticles into packaging materials is a nontrivial process, dramatically reducing the performance of the nanostructures because of agglomeration [30]. Thus, direct production of nanostructures on packaging materials is desirable. Magnetron sputtering is a candidate method to achieve this, since it is commercially used in packaging materials and allows precise control of the morphology and composition of the nanostructures, reducing the use of toxic chemicals during processing [19,31]. 

In the present work, four different coating systems with dissimilar atomic arrangements, namely pure Zn, Zn-Fe alloy, Zn-Fe nanolayers (Zn-Fe NLs), and Zn nanolayers containing Fe nanoparticles (Zn NLs + Fe NPs), were produced. Conventional magnetron sputtering and hybrid magnetron sputtering coupled to a cluster gun were used to control the morphology, structure, and chemical composition of the coatings, which were later correlated with the coatings’ antibacterial and chromatic properties. It was demonstrated that the design of the coatings’ architecture allowed control of the oxidation of the Zn-Fe nanostructures, leading to controllable changes in their chromatic and antibacterial properties. Finally, the system with the best functionalities to be used in food packaging materials was established.

## 2. Materials and Methods

### 2.1. Production of the Materials

The nanostructures were produced using two different sputtering processes to adjust the atomic arrangements between the Zn and Fe, as shown in Figure 1. Classical magnetron sputtering was used to produce pure Zn, Zn-Fe alloy, and Zn-Fe nanolayers (Zn-Fe NLs) coatings as shown in Figure 1a, b, d, respectively. On the other hand, a hybrid magnetron sputtering with a cluster gun was used to produce the Zn nanolayers containing Fe nanoparticles (Zn NL + Fe NPs) coating system, as shown in Figure 1c.

#### 2.1.1. Classical Magnetron Sputtering

Figure 2a presents a top-view layout of the classical chamber where the samples were produced. The nanostructures were deposited using a DC-pulsed magnetron sputtering technique with two 200 × 100 mm^2^ high-purity targets of Zn and Fe located 180° from each other (Zn TRG, 99.99%, Fe TRG, 99.95%, acquired from Testbourne Ltd., Basingstoke, UK). The chamber was evacuated at an initial pressure of 2 × 10^−4^ Pa, then set up to a working pressure of 5 Pa by introducing argon (ΦAr = 80 sccm). The substrates, rotating at a constant speed of 8 rpm, were maintained at a constant temperature between 300 and 313 K during the deposition.

The deposition parameters are presented in Table 1. Because of the low Fe deposition rate, the process was performed by turning the Zn target ON/OFF while the Fe target was always maintained at ON (Figure 2b), which allowed for better control of the composition of the coatings. Zn-Fe alloy coatings, for instance, were deposited in 15 thin layers to ensure the alloy morphology by distributing Zn and Fe over the entire surface, while for Zn-Fe NLs, the Zn target was turned OFF only at the end of the Zn layer deposition to allow the bilayer morphology.

#### 2.1.2. Hybrid Magnetron Sputtering—Cluster Gun

Figure 2c shows the scheme of the chamber setup where the Zn NL + Fe NP deposition was performed. It was divided into two parts: (i) the main chamber, where the Zn target (diameter 50.8 mm, thickness 4.5 mm, and purity: 99.9%, purchased from Testbourne Ltd., Basingstoke, UK) was placed at 6 cm from the substrate, and (ii) the gun chamber, where the Fe target (diameter 6.9 cm, thickness 3 cm, purity 99.95% obtained from Testbourne Ltd., Basingstoke, UK) was located. The gun chamber contained a DC magnetron sputtering cluster source and was connected to a water-cooling system. This chamber had two apertures with diameters of 2.5 mm and 4 mm, respectively, to guarantee that the flow direction of the cluster beamed toward the main chamber because of pressure differences. The substrate holder was located 10 cm from the aperture, while the Fe magnetron was 8 cm away from the large nozzle. Ar was used as a sputtering gas with the flow ΦAr = 60 sccm, resulting in working pressures of 88 Pa and 0.4 Pa in the cluster source and the main chamber, respectively. 

The morphology of the Zn NLs + Fe NPs (Figure 1c) was produced using alternate deposition by turning ON/OFF the power applied to each magnetron (as shown in Figure 2d). This multilayer system allowed for precise control of the content of Fe because of the lower deposition rate of Fe in the cluster gun compared with the conventional magnetron sputtering used for Zn deposition. An example of the final coating 

All coatings were deposited onto Si-wafers (supplied by Siegert Wafer GmbH, Aachen, Germany), TEM Cu-grids with ultrathin carbon layers (400 mesh, obtained from Monocomp Instrumentación S.A., Madrid, Spain), transparent glass slides (purchased from Fisher Scientific, Leicestershire, UK), and poly L lactic acid (PLA) biopolymer films (50 µm thickness, acquired from Goodfellow GmbH, Hamburg, Germany). The Si and glass substrates were sequentially cleaned with distilled water, acetone, and 95% ethanol (10 min with each solvent) in an ultrasonic bath to remove impurities on the surface, while the PLA and the Cu-grids were placed as bought because of their sensitivity to solvents.

### 2.2. Methodology

#### 2.2.1. Morphology, Composition, and Structure

Scanning electron microscopy (SEM) images of the coatings deposited on Si-wafers were performed with an FEI Helios NanoLab 450S Dual Beam (Eindhoven, The Netherlands) with a through-the-lens detector (TLD), operating at 5 keV with a beam current of 0.4 nA. High angle annular dark-field (HAADF) images obtained by scanning transmission electron microscopy (STEM) were collected from samples deposited on ultrathin carbon grids at 200 keV on a double corrected FEI Titan Themis (Eindhoven, The Netherlands). Furthermore, energy-dispersive X-ray spectroscopy (EDS) mapping in transmission mode was acquired using a double corrected FEI Titan Themis (Eindhoven, The Netherlands) operated at 200 keV and equipped with a Super-X EDS detector. To determine the elemental distribution, iterative maps of 512 × 512 pixels with a dwell time per pixel of 10 µs at 200 keV were acquired. 

Inductively coupled plasma optical emission spectroscopy (ICP-OES) was used to quantify the total concentration of Zn and Fe in each coating deposited on glass slide substrates. Thus, samples of approximately 1 cm^2^ were placed in 15 mL Falcon tubes and immersed in 10 mL of 65% (*v*/*v*) HNO_3_. The samples were left at room temperature for 24 h and then diluted and filtrated through 0.22 µm of regenerated cellulose membranes. The measurements were carried out in an Optima 8000 ICP-OES (Perkin Elmer, Boston, MA, USA), operating with an axial plasma view and 1400 W, at wavelengths of 214 and 238 nm for Zn and Fe, respectively.

X-ray photoelectron spectroscopy (XPS) of the coatings deposited on PLA films was performed in a Thermo Scientific ESCALAB 250Xi (Eindhoven, The Netherlands) with the aim to determine the surface composition and chemical binding energies. The analysis was conducted at 15 kV (200 W) with a monochromatic Al Kα X-ray source of 1486.7 eV. Data acquisition was carried out with a charge neutralization system and a pressure lower than 1 × 10^−6^ Pa. 

Additionally, structural analysis was performed to identify the phases in the coatings. Selected area electron diffraction (SAED) patterns were obtained using samples deposited on ultrathin carbon grids.

#### 2.2.2. Functional Properties

Color parameters and the opacity of the coatings deposited on glass substrates were determined using a Minolta colorimeter (CR 400, Minolta, Japan). The equipment was calibrated with a white standard color plate, used as a background for color measurements (*L**, *a**, *b**) according to Ballesteros et al. [32]. The opacity of the coatings, expressed in percentage (%), was calculated by the Hunter lab method, using the ratio of the opacity of each sample on a black standard (*Yb*) and a white standard (*Yw*), as described by Casariego et al. [33]. The results were simulated using Adobe Photoshop software (CS6, Adobe Inc. San Jose, CA, USA)). Five replicates of each coating were made for both color and opacity measurements, which were analyzed as deposited (1st day) and after being exposed at 98% relative humidity (7th day) using a saturated salt solution of K_2_SO_4_ at room temperature.

The antibacterial activity of the coating studied was performed against the Gram-positive *Staphylococcus aureus* (EG17 strain) bacterium and Gram-negative *Escherichia coli* (CECT 736 strain) bacterium, obtained from the Centre of Biological Engineering collection of the University of Minho. The zone of inhibition (ZOI) test was carried out as described by the Clinical and Laboratory Standards Institute [34] in order to determine the diffusion in the agar of Zn and Fe from the PLA film’s surface, which was used as a substrate. The halo size was used to quantify the inhibition area of the coatings against bacterial growth. The bacteria were cultivated into 20 mL of nutrient broth (NB, Oxoid) and incubated at 37 °C, 150 rpm for 18 h. The resultant cell suspension for each strain was adjusted to an optical density (80–82% in the McFarland standards) between 0.09 and 0.11, measured at 620 nm, indicating a concentration of 1 × 10^8^ CFU/mL. Later, the inocula were diluted in NB to 1 × 10^6^ CFU/mL, and then, aliquots of 200 µL of cell suspensions were spread with sterile swabs on Petri dishes (90 mm) containing approximately 25 mL of nutrient agar (NA, Oxoid). The Zn and Fe coatings deposited on PLA films of approximately 0.5 × 1.5 mm^2^ were previously sterilized by exposing them to UV light for 1 h and subsequently placed in contact with the agar. The Petri dishes were incubated at 37 °C for 24 h, and pure PLA films were used as a negative control. The transparent halo formed around the samples was evaluated for each bacterium to define the inhibition of bacterial growth. The ZOI was measured through the Image J-64 software (V1.52p, National Institute of Health, Bethesda, MA, USA) and expressed in mm^2^. Each sample was evaluated in duplicate and repeated at least in two independent assays.

#### 2.2.3. Statistical Analysis

GraphPad Prism V6.1 by Dotmatics (San Diego, CA, USA) was used to carry out a one-way analysis of variance (ANOVA) and Tukey’s multiple comparisons test to evaluate the significant differences (*p* < 0.05) among the different coatings.

## 3. Results and Discussion

### 3.1. Morphology, Structure, and Chemical Composition Characterizations

STEM-EDS chemical mapping was carried out to confirm the morphologies and element distribution of the Zn-Fe nanostructures, as shown in Figure 3. Both a top view and a cross-section were acquired. The results showed different Fe distributions depending on the production method. For comparison, a pure Zn nanostructure was shown, revealing a coating formed by large grains with random shapes (Figure 3a). The Zn-Fe alloy showed a homogeneous distribution, as depicted in Figure 3b. On the other hand, the Zn NL + Fe NP sample (Figure 3c) showed the presence of Fe NPs, which was due to the agglomeration process that occurred in the cluster gun. These NPs had a large size distribution, forming a separate phase that was easily identified. The Fe NPs varied between 5 and 23 nm, with an average size of 11 nm (see Appendix A). Finally, Figure 3d shows that the Zn-Fe NL sample had a bilayer morphology and a uniform distribution of Fe surrounding the Zn nanolayer. All the coatings exhibited passivated surface and column boundaries, as observed in the oxygen signal, in agreement with the XPS analysis, as later demonstrated.

Lower magnification STEM top-view images and SEM cross-section images are shown in Figure 4. The results evidenced full coverage of the samples’ surface, but with significant differences in the morphology depending on the coatings’ architecture. All the coatings revealed a large distribution of grain sizes (Figure 4a–c), except for the Zn-Fe alloy, which showed a more compact morphology (Figure 4b) than all the other samples. Both the Zn NLs + Fe NPs and Zn-Fe NLs had heterogeneous particle distributions, but the latter showed a clear bimodal particle size distribution. 

Table 2 shows that the Zn at.% was approximated (81–86 at.%) for most of the samples, except for the Zn-Fe alloy coating, which featured almost half of this composition (46 at.%). The Zn-Fe NL and Zn NL + Fe NP nanostructures presented very close Fe at.% values, while Zn-Fe alloy, as expected, had a higher amount of Fe. The O at.% was lowest for the Zn-Fe NLs (4 at.%), similar for the pure Zn and Zn NLs + Fe NPs (14 and 11 at.%, respectively), and around 30 at.% for the Zn-Fe alloy. It is worth noting that in a previous study, Zn-Fe alloys with 9 at.% of Fe showed passivation against the oxidation of the coating [23]. Thus, in the present investigation, the Zn-Fe alloy sample was produced with a higher amount of Fe to promote the oxidation of the film for the expected chromatic and antibacterial properties. The presence of oxygen in all the samples was ascribed to a spontaneous oxidation reaction when the small metal nanostructures were in contact with air.

The total concentration of Zn and Fe of the samples is also shown in Table 2. The sample with the highest Zn concentration was the Zn-Fe NLs, followed by the Zn NL + Fe NP, Zn-Fe alloy, and pure Zn coatings. Although the thicknesses of the Zn-Fe NL and Zn NL + Fe NP coatings were similar, the different production methods led to modifications in the morphologies and therefore in the concentrations. 

XPS depth profiles were carried out to study the chemical bonding of the coatings as a function of the thickness for Fe 2p, O 1s, and Zn LMM. It is important to highlight that the X-ray-induced Zn LMM Auger peaks had a larger shift with the chemical state than those of metallic Zn and ZnO, and therefore, this was preferred over Zn 2p. Figure 5 shows the evolution of the Zn, Fe, and O as a function of the argon sputtering time, showing that the coatings were composed of a metallic and oxidized mixture of Zn and Fe.

Only a fraction of the Zn and Fe was oxidized, which accounted for the passivation of the metals at the surface already observed in the STEM images. The pure Zn sample (Figure 5a) also showed a combination of Zn/ZnO, which was attributed to the passivation of Zn in a natural environment. The incorporation of Fe in the coatings intensified the oxidation of Zn, as evidenced by the intensity ratio between the ZnO peak located at 988 eV [34] and the metallic Zn peak at 992 eV [35], as shown in Figure 5b–d. This effect was likely potentiated by the galvanic couple created by the Zn and Fe. The oxidation of Zn and Fe was much more pronounced in the Zn-Fe alloy coating (Figure 5b). Furthermore, the three samples with Zn and Fe (Figure 5b–d) contained Fe in both metallic oxidized states. The components located at 706 and 720 eV were attributed to Fe-Fe, whereas those at 711 and 724 eV were attributed to Fe^3+^ in Fe_2_O_3_ [36,37]. Note that no significant shifts in the peak location were observed as a function of thickness or etching time.

All samples revealed two peaks in the O 1s, with different intensity ratios. The first peak was located at 530.7 eV, related to the O atoms in the metal oxides. The second, at 532 eV, was attributed to O in oxygen-deficient regions within the matrix of ZnO, which explains the changes in the intensity of this peak for all the samples, since variation in the concentration of oxygen vacancies were expected [38]. The XPS O 1s patterns of the Zn NL + Fe NP coating (Figure 5c) were most intense, which was due to the high amount of O that was already noticed with the at.% of oxygen in Table 2. Furthermore, the presence of the oxygen peak at higher energies for the final etching time corresponded to the C-O and C=O for PLA.

As shown in Figure 6, selected area electron diffraction was acquired from samples observed from the top view, showing the presence of polycrystalline materials for pure Zn (Figure 6a), Zn NL + Fe NP (Figure 6c), and Zn-Fe NL (Figure 6d) coatings. However, the Zn-Fe alloy (Figure 6b) exhibited quasiamorphous phases but with distinctive rings. The pattern profiles of the samples are overlaid in Figure 6. The peaks of pure Zn matched well with a mixture of Zn and ZnO phases, both hexagonal. More diffused rings were observed in the coatings with the incorporation of Fe, indicating smaller crystals in the Zn-Fe coatings due to the Fe addition. No evidence of additional phases was observed in the Zn-Fe samples, despite the clear existence of isolated Fe layers observed in the cross-section images. In the Zn-Fe alloy coating, two large peaks were observed, which are indistinguishable from the Zn, ZnO, or Fe_2_O_3_ phases.

### 3.2. Functional Properties

#### 3.2.1. Chromatic Properties

Figure 7a, b shows the color and opacity of the coatings, respectively, before and after exposing them to a high relative humidity (RH) environment for 7 days (RH = 98%) with the aim to promote the oxidation of the metallic nanostructures.

In Figure 7a, the bar plot indicates the lightness (*L**) of the coatings in the *L**, *a**, *b** CIELAB color space. The colors of the bar were selected to show the real color of the samples using the calculated RGB colors converted from the CIELAB color space. The *L** for the four as-deposited samples varied from 35 to 90, revealing a correlation with the metallic content of the coatings. After seven days in a high RH environment, the *L** of the pure Zn and Zn-Fe NL coatings increased significantly, while that of the Zn-Fe alloy and Zn NL + Fe NP samples marginally decreased from the value measured on the first day. 

The opacity results of the coatings on the first day and after 7 days at 98% RH are shown in Figure 7b. The bars are filled by the calculated color combined with the calculated opacity. According to the histogram, the opacity decreased significantly for all samples produced by a classical magnetron (pure Zn, Zn-Fe alloy, and Zn-Fe NLs), while it did not change for the Zn NL + Fe NP coating. It is worth noting that the Zn was the predominant phase in the samples; thus, it could be the element responsible for the opacity change. During the oxidation, ZnO, which is a known transparent conductive oxide [39], was formed. The calculated RGB color models before and after exposing the coatings to a high RH were different for all the samples except the Zn NL + Fe NP coating (produced by the cluster gun), which had very similar results for both measurements.

The high change in color, lightness, and opacity for the Zn-Fe NL coating was attributed to its morphology. The outer thin Fe layer was formed of small Fe grains (<2 nm), which was expected to offer high chromatic influence by changing colors during the oxidation reaction, leading to simultaneous oxidation of Zn and Fe. Thus, Zn-Fe NLs presented the highest decrease in opacity, from 100 to 10%, indicating high production of ZnO at high humidity and proving the effect of Fe (in this morphology) on the promotion of oxidation when compared with pure Zn coatings (94 to 25%). 

In the Zn NL + Fe NP sample, the Fe was located as clusters distributed in the different zones of the coating. Consequently, the oxidation was expected to occur sporadically with different rates (at higher rates where Zn was in contact with Fe and slower rates in pure Zn spots). As shown in Figure 3c, Zn and Fe were deposited in separate phases, and the oxygen was more concentrated at the boundaries of the clusters (see cross-section images). This morphology limited the oxidation reaction and well protected the metallic particles. As a result, an absence of chromatic properties was observed for the Zn NLs + Fe NPs.

On the other hand, it is known that Zn-Fe alloys are used to increase the corrosion resistance in wet environments [40], as was already confirmed in a previous study by our group in which an alloy sample with 9% of Fe showed passivation against the oxidation of the coatings [23]. This explains the weak color change in the coating of Zn-Fe alloy, indicating a slow oxidation process. However, the weakness of the chromatic properties for this sample indicated that the compact morphology (Figure 3b) prevented access to oxygen, increasing the oxidation resistance.

#### 3.2.2. Antimicrobial Properties

The antimicrobial properties of the nanostructures containing Zn and Fe are presented in Figure 8a. All the coatings were tested against the Gram-negative bacterium *E. coli* and the Gram-positive bacterium *S. aureus*. For *S. aureus*, the Zn-Fe NL and Zn NL + Fe NP samples showed a higher inhibition than the pure Zn and Zn-Fe alloy samples. The Zn-Fe alloy sample did not present inhibition against *E. coli*, in contrast to pure Zn, which showed the greatest effect, followed by the Zn-Fe NL and Zn NL + Fe NP coatings. The areas of inhibition for both bacteria were similar when the pure Zn coating was tested, while it was higher for *S. aureus* with the Zn-Fe NL and Zn NL + Fe NP coatings. The Zn-Fe alloy showed no activity against *E. coli*. These results evidence a complex interaction between different factors on the antibacterial properties, which are explained below:(i)Effect of Zn-Fe concentration

ZnO is a well-known antibacterial agent, and thus, it was expected to trigger the antibacterial activity in the produced nanostructures, since Zn was their major constituent. In addition, increasing Zn concentration in the samples (Figure 8b) was expected to proportionally raise the antibacterial effect. The test against *S. aureus* agreed with this. The nanostructures with more Zn content (Zn-Fe NLs and Zn NL + Fe NPs) showed the highest inhibition against this bacterium, but without an obvious influence of the Fe concentration on the antibacterial activity of the coatings. On the other hand, the coatings showed different behavior against *E. coli*. Pure Zn coating (Figure 8a), for example, exhibited the highest antibacterial action against this bacterium despite reflecting the lowest content of Zn, suggesting that Fe exhibited a negative impact on the effect of Zn against *E. coli*.

(ii)Effect of Fe concentration

When Fe was incorporated into the coating, a reduction in or complete absence of antibacterial activity was observed against *E. coli*. Some studies have shown that the presence of ferric oxide as a nutrient or as an electron acceptor in water under anaerobic conditions increases the bacterial cultivability, enhancing the growth of *E. coli* [41]. 

As a result, the absence of antibacterial activity of Zn-Fe alloy coating against *E. coli* could be related to the high content of Fe (Figure 8b and Appendix A) in this sample, which could counteract the antibacterial effect of ZnO against this Gram-negative bacterium. On the other hand, the Zn NL + Fe NP and Zn-Fe NL coatings showed low inhibitions against *E. coli*, but with significant differences between them (Figure 8a), despite both possessing high Zn content and similar Fe content. This behavior could be attributed to the Zn and Fe distributions in the coatings. 

(iii)Effect of Zn and Fe distribution

The morphology and Zn-Fe distribution may influence the antibacterial activity of the samples because of the availability of Zn and Fe in the nanostructures. The distinctive behavior of the Zn-Fe NL coating against *E. coli* was attributed to the fact that Fe formed a continuous layer, which may have made its absorption by the Gram-negative bacteria difficult, while in the Zn NL + Fe NP and Zn-Fe alloy coating systems, the Fe atoms were more readily available for the bacteria to absorb.

In general, the antibacterial data agreed with the oxidation of the samples, which was monitored by the color changes of the coatings. The two samples with higher chromatic changes (pure Zn and Zn-Fe NLs) also showed enhanced antibacterial activity, presenting the most inhibition against both *S. aureus* and *E. coli* bacteria. Therefore, both antibacterial activity and color were tightly linked to the oxidation of the samples, since the coatings that continued the oxidation process, after contacting high relative humidity environments, were expected to change color and exhibit antibacterial effects. On the contrary, very stable samples, such as Zn-Fe alloy, presented small color and opacity changes as well as low inhibition against the growth of *S. aureus* and no effect against *E. coli.*

## 4. Conclusions

Conventional magnetron sputtering and hybrid magnetron sputtering with a cluster gun were used to produce Zn-Fe nanostructured systems with different architectures and chemical compositions, which in turn influenced the active and intelligent properties of the produced materials. STEM and SEM results revealed morphological differences among all the nanostructures, independently of the production method employed. XPS and SAED data of deposited coatings showed the Zn and Fe phases in their metallic states beside oxide phases. All the coatings presented significant antibacterial activity against *S. aureus*, revealing a direct correlation with the Zn content in each sample. On the other hand, the pure Zn coating showed the highest inhibition area against *E. coli*, while Fe presented a negative impact on inhibiting the growth of this bacterium, decreasing the effect to less than half the zone of inhibition for Zn-Fe NLs and Zn NLs + Fe NPs and suppressing the antibacterial effect for the Zn-Fe alloy coating when compared with the pure Zn coating. Finally, the Zn NL + Fe NP and pure Zn coating systems exhibited interesting chromatic properties, with reductions in opacity from 100 to 10% and 93 to 23%, respectively, confirming the oxidation of the nanostructures when exposed to high humidity environments. Therefore, this study allows concluding the great potential of the coatings produced to be used as active and intelligent food packaging and supports directing further research and development into the packaging field.

## Figures and Tables

**Figure 1 nanomaterials-12-02104-f001:**
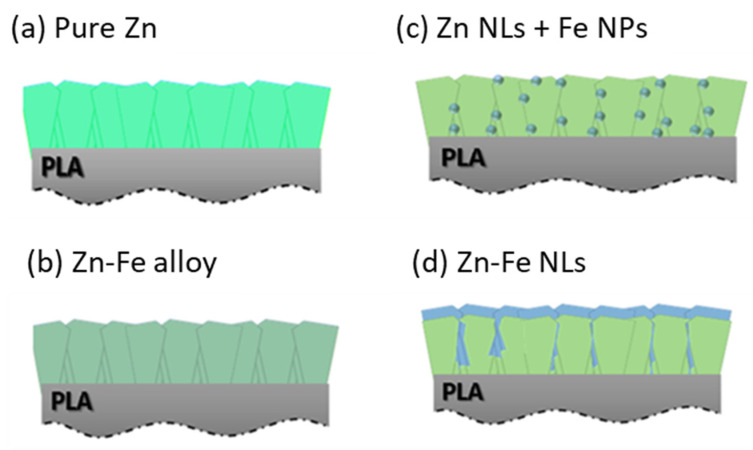
Diagram of the different morphologies produced by classical magnetron sputtering and hybrid magnetron sputtering with a cluster gun: (**a**) pure Zn; (**b**) Zn-Fe alloy; (**c**) Zn NLs + Fe NPs; and (**d**) Zn-Fe NLs.

**Figure 2 nanomaterials-12-02104-f002:**
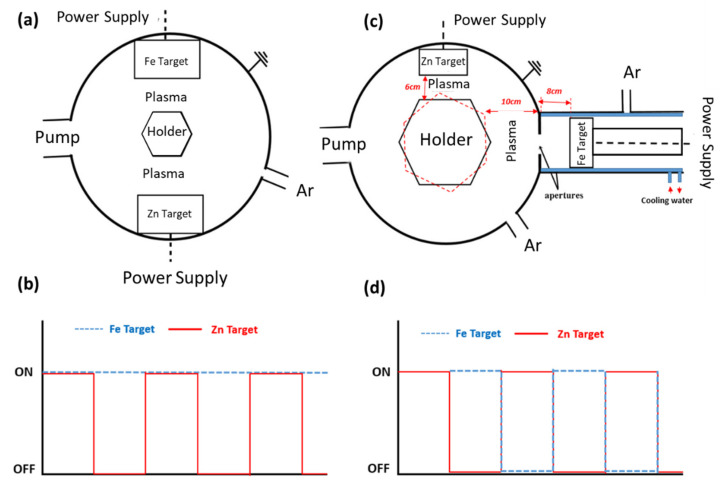
(**a**) Top-view layout for classical magnetron sputtering; (**b**) Zn and Fe target ON/OFF state during classical magnetron deposition; (**c**) top-view layout for hybrid magnetron sputtering with a cluster gun; and (**d**) Zn and Fe target ON/OFF state during hybrid deposition.

**Figure 3 nanomaterials-12-02104-f003:**
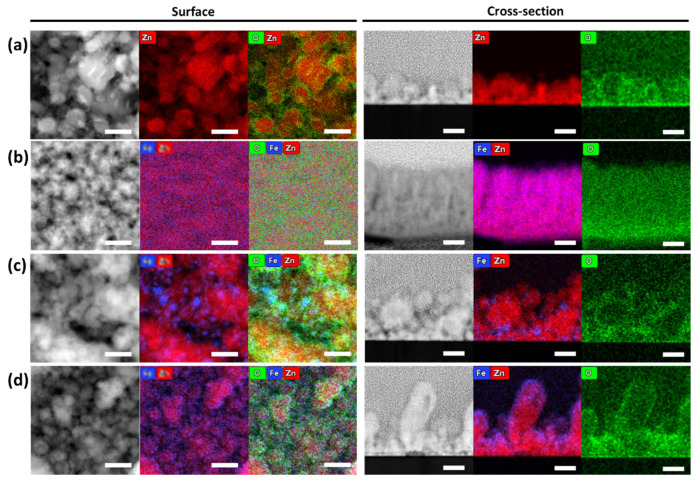
STEM-EDS images of the coatings: (**a**) pure Zn; (**b**) Zn-Fe alloy; (**c**) Zn NLs + Fe NPs; and (**d**) Zn-Fe NLs, viewed from the top (left) and cross-section (right). The scale bar corresponds to 40 nm. The red, blue, and green colors in the figure correspond to Zn, Fe, and O signals, respectively.

**Figure 4 nanomaterials-12-02104-f004:**
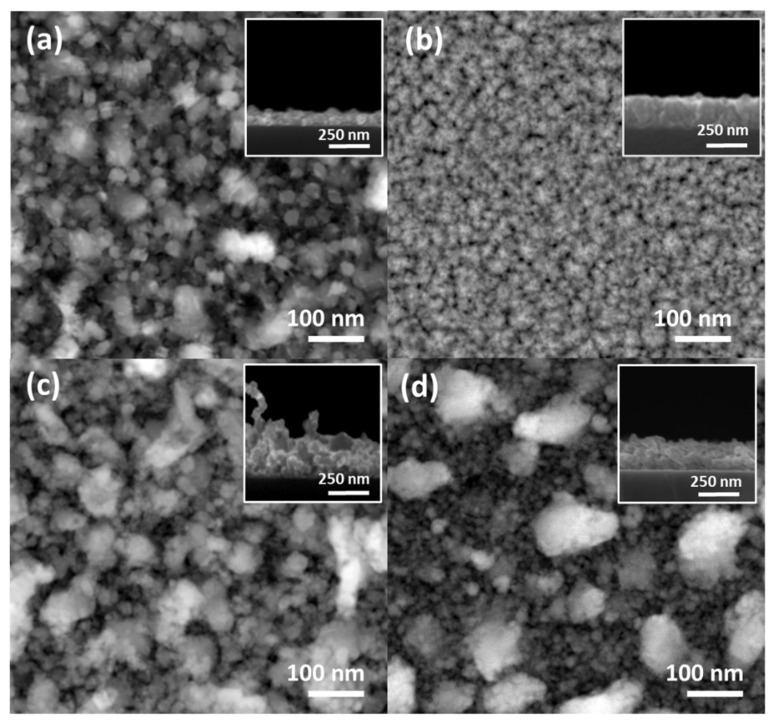
SEM surface images of the coatings: (**a**) pure Zn; (**b**) Zn-Fe alloy; (**c**) Zn NLs + Fe NPs; and (**d**) Zn-Fe NLs. The inset in each image represents the SEM cross-section for each sample.

**Figure 5 nanomaterials-12-02104-f005:**
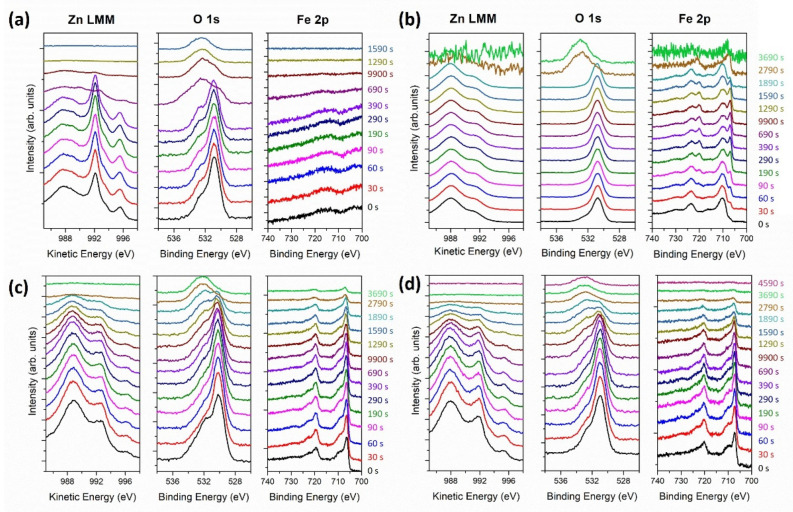
XPS spectra of the coating as a function of sputtering time: (**a**) pure Zn; (**b**) Zn-Fe alloy; (**c**) Zn NLs + Fe NPs; and (**d**) Zn-Fe NLs.

**Figure 6 nanomaterials-12-02104-f006:**
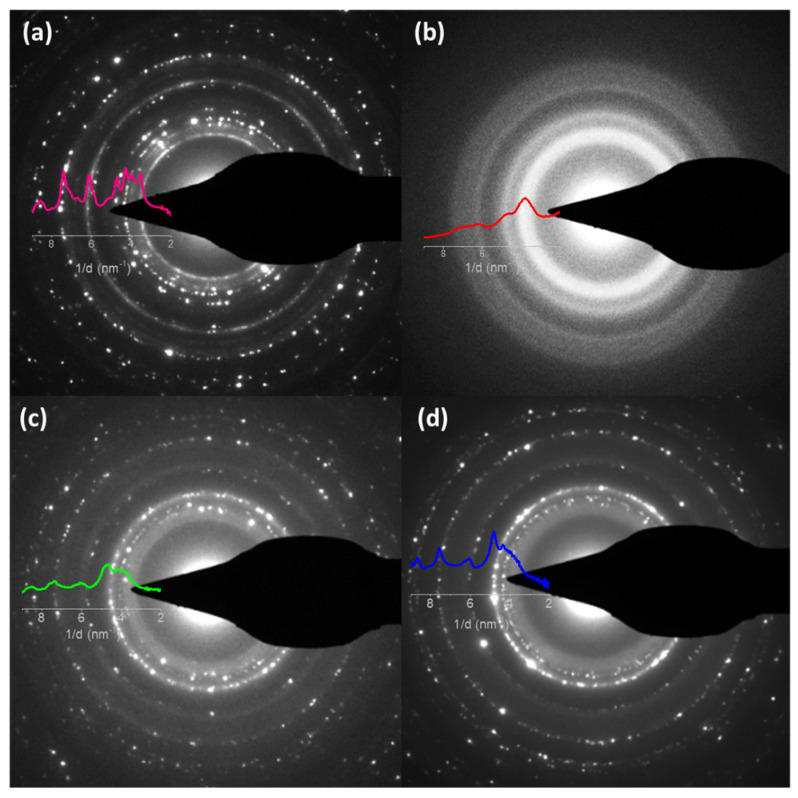
SAED images obtained for each coating: (**a**) pure Zn; (**b**) Zn-Fe alloy; (**c**) Zn NLs + Fe NPs; and (**d**) Zn-Fe NLs.

**Figure 7 nanomaterials-12-02104-f007:**
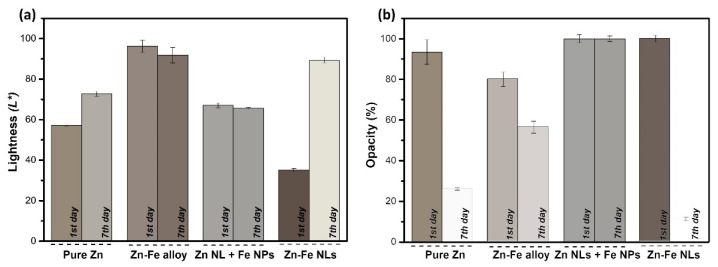
(**a**) Lightness (*L**) values and (**b**) opacity percentages of the systems of pure Zn, Zn-Fe alloy, Zn NLs + Fe NPs, and Zn-Fe NLs being analyzed as deposited (1st day) and after being exposed at 98% relative humidity (7th day). Low *L** values correspond to dark samples, while high *L** values belong to light samples. The color of the bar represents the real color of the samples as simulated by RGB color model, not including the opacity in (**a**) and including the opacity values in (**b**).

**Figure 8 nanomaterials-12-02104-f008:**
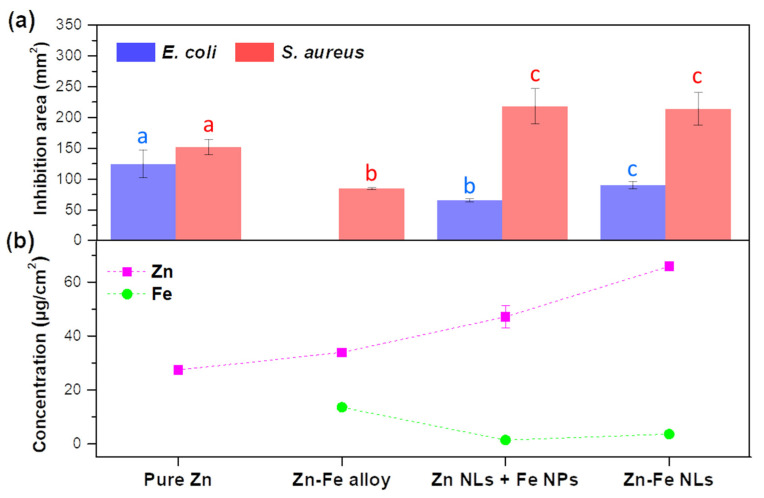
(**a**) Inhibition area of the coating systems, pure Zn, Zn-Fe alloy, Zn NLs + Fe NPs, and Zn-Fe NLs, against *E. coli* and *S. aureus* (different letters within each bacterium mean values were statistically different at 95% confidence level); (**b**) Zn and Fe concentrations in the different coatings as obtained by ICP-OES.

**Table 1 nanomaterials-12-02104-t001:** Deposition parameters to produce the different coating systems.

Process	Coating	J_Zn_ (mA/cm^2^)	J_Fe_ (mA/cm^2^)	Zn + Fe Layer Time (min)	Fe Layer Time (min)	N. of Layers
Classical	Pure Zn	0.5	—	14.5 *		1
Classical	Zn-Fe alloy	0.5	2.5	1	4.17	15
Cluster	Zn NLs + Fe NPs	1.9	3.2	0.5*	10	24
Classical	Zn-Fe NLs	1.0	2.5	15	10	1

*** only Zinc target is active; J: current density.

**Table 2 nanomaterials-12-02104-t002:** Experimental details used during deposition, thickness, atomic composition, and final mass concentration of Zn and Fe in the produced nanostructures.

Deposit Conditions					Characteristics of the Produced Coatings					
**Coatings**	Deposition time (min)	Current Density (J) (mA/cm^2^)		Deposition Rate (nm/s)	Thickness (nm)	Atomic Percent (at.%)			Metal Concentration by ICP-OES (µg/cm^2^)	
		J_zn_	J_Fe_			Zn	Fe	O	Zn	Fe
**Pure Zn**	Zn = 14.5	0.5	—	0.13	109	86	—	14	27.40 ± 0.43	—
**Zn-Fe alloy**	Zn = 15 Fe = 62.5	0.5	2.5	0.30	175	46	24	30	33.90 ± 0.60	13.51 ± 0.97
**Zn NLs + Fe NPs**	Zn = 12 Fe = 240	2.5	3.2	0.02	238	81	8	11	47.13 ± 4.21	1.40 ± 0.52
**Zn-Fe NLs**	Zn = 15 Fe = 25	1 *	2.5 *	0.09	207	89	7	4	69.20 ± 1.34	3.77 ± 0.11

* Zn and Fe simultaneously deposited.

## Data Availability

Not applicable.

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
