# Peer review of "Zn and Zn-Fe Nanostructures with Multifunctional Properties as Components for Food Packaging Materials"

_nanomaterials, 2022, doi:10.3390/nano12122104_

Round 1

Reviewer 1 Report

The article contains a comprehensive study of the structure (architecture type) of Zn-Fe depending on the functional properties. The work uses a sufficient number of complementary research methods. In general, the article is written at a high professional level. However, in terms of XPS, I would like to get some changes:

1) the etching time on the spectra is not signed;

2) it is not clear why there is a peak in the high-energy region of the oxygen spectrum after etching, since judging by the energy, it refers to C=O bonds;

3) there is no information on the concentration of elements. It seems to me that this information should be added to Table 3, which should also include the peak positions of Fe2p, ZnLMM, and O1s for at least 3 etch times (beginning, middle, end).

In addition, I wanted to clarify how the oxygen concentration in the Zn-Fe structure affects the antibacterial properties.

The data obtained from Figures 3c and 4c do not match.

Author Response

The article contains a comprehensive study of the structure (architecture type) of Zn-Fe depending on the functional properties. The work uses a sufficient number of complementary research methods. In general, the article is written at a high professional level. However, in terms of XPS, I would like to get some changes:

The etching time on the spectra is not signed

Response: The etching time was added to the Figure (Please see the modified Figure 5 in the revised manuscript)

2) it is not clear why there is a peak in the high-energy region of the oxygen spectrum after etching, since judging by the energy, it refers to C=O bonds;

Response: We appreciate the reviewer comment. It was something that we carefully check before during the analysis of the results. The samples revealed two peaks in the O 1s, with different intensity ratios. The first peak is located at 530.7 eV is related to the O atoms in the metal oxides. Whereas the presence of the second peak at 532 eV is attributed to O in oxygen-deficient regions within the matrix of ZnO, which explains the changes in the intensity of this peak for all the samples since variation in the concentration of oxygen vacancies is expected.

In addition, after complete etching of the films, the signal of oxygen belongs to the substrate (PLA), a sentence clarifying the presence of that peak was added to the manuscript. (Please see Lines 326 – 327 in the revised manuscript).

3) there is no information on the concentration of elements. It seems to me that this information should be added to Table 3, which should also include the peak positions of Fe2p, ZnLMM, and O1s for at least 3 etch times (beginning, middle, end).

Response: The concentration of the elements was evaluated by EDS and ICP-OES. We avoid the calculation of the concentration of the Zn and Fe in the XPS results due to the clear difference in sputtering rates of Zn and Fe, which will mislead the reader. In addition, no significant shifts are observed during etching, thus the position of the peaks reported in the manuscript are valid for all the etching times. A note clarifying this was added to the manuscript (Please see Lines 317 – 318 in the revised manuscript).

In addition, I wanted to clarify how the oxygen concentration in the Zn-Fe structure affects the antibacterial properties.

Response: We thank the reviewer for the question. As explain in the text, the passivation of the films is a significant event that prevents the change in color to occurs immediately after the production, which allows the storage of the samples for large periods of time. As soon as they are in contact with high relative humidity, that protective layer is not longer very effective and allow the oxidation of the metals. In the case of the antibacterial tests, it has been widely reported that the Zn+2 ions are the responsible for the antibacterial effect in case of ZnO and thus, the dissolution of the oxide is required to obtain an antibacterial effect. If the samples are only partially oxidized, the non-oxidized part of the material, in the interior of the films, could be access more readily to act as antibacterial agent. In fact, a deterioration of the antibacterial properties is observed for the Zn-Fe alloy where the oxidation is more uniformed. However, other parameters such as morphology and Zn concentration must not be ignored.

A sentence about the oxygen effect on the antibacterial effect was modified to clarify this point (Lines 449 – 457 in the revised manuscript), as follows:

“In general, the antibacterial data agrees with the oxidation of the samples, which is monitored by the color changes of the coatings. The two samples with higher chromatic changes (pure Zn and Zn-Fe NLs) also show an enhanced antibacterial activity, presenting the most inhibition against both S. aureus and E. coli bacteria. Therefore, both antibacterial activity and color are tightly linked to the oxidation of the samples since the coatings that can readily continue the oxidation process after contact with high relative humidity environments, are expected to both change color and exhibit antibacterial effects. On the contrary, very stable samples, such as Zn-Fe alloy, present small color and, opacity changes as well as low inhibition against the growth of S. aureus and no effect against E. coli.”

The data obtained from Figures 3c and 4c do not match.

Response: We are not completely sure what the reviewer means by not matching the results. The images present similar grain sizes, but due to the inhomogeneity of that sample, the surface and cross-section view significantly depend on the region of observation. That is the reason why we also include lower magnification SEM data. It is important to highlight that the information of the SEM cross-section is acquired by fracturing a Si substrate and observing the cross-section. In that method, we observed a projection of the grains in a thick sample, much thicker than for a TEM sample. In the TEM observation, we prepared the sample using FIB, obtaining thickness values of around 60 nm.  That methodology allows us to observe a couple of grains in projection.

Reviewer 2 Report

This manuscript produces Zn-Fe nanostructures with different morphologies: i) nano-alloys; ii) nano-layers; and iii) nano-layers combined with nanoparticles, aiming to produce materials with multifunctional properties. The idea is good and sounded; however, many issues should be addressed according to the following comments:

1) The overall presentation, readability, and more results and analysis are mandatory. Please, correct the language problems, it is weak from the Grammarly and sequences of events, I catch 22 errors by using a personal program, and the authors should cure them carefully.

2) The "Abstract" section should be more intensively focused on the main idea directly and must contain the contribution of this manuscript supported with numerical result indicators. Also, please write the abbreviations of all nanoparticles " Zn and Fe" in the abstract section.

3) The "Introduction" section should be made much more impressive by highlighting your contributions. The novelty of this manuscript must be explained simply and clearly in points at the end of the introduction section. Note that, the introduction section should consist of three parts, i.e., a general introduction to the topic, followed by a literature survey, then the contribution clarifications.

4) The introduction section should be enriched with up-to-date references by adding and citing the latest trends in the area of the influence of the nanoparticle's functionalization process on the morphology, structure, and characterization within the food packaging materials matrix. E.g., Effect of functionalized TiO2 nanoparticles on dielectric properties of PVC nanocomposites & Recent Advances in Polymer Nanocomposites Based on Polyethylene and Polyvinylchloride for Power Cables & PVC Nanocomposites for Cable Insulation with Enhanced Dielectric Properties, PD Resistance & Development of Industrial Scale PVC Nanocomposites with Comprehensive Enhancement in Dielectric Properties.

5) It is mandatory to check carefully all the abbreviation definitions, symbols, and standard units in the whole manuscript. I catch some errors and the other symbols are not defined, please, define the abbreviations. Also, there are many standard tests without citing their references.

6) The resolution and quality of figures must be modified; they should be presented as close to the camera-ready format (e.g., Figs. 2, 5, and 7). Also, please don't use the symbol abbreviations on X-Y-axes, they must have the full name with their SI units.

7) It will be helpful to the readers if a new subsection 3.3 is added that contains various discussions about the obtained results and other comparisons with previous studies.

8) The conclusion section should be more concentrated and supported by the numerical results. Also, the authors may propose some interesting problems as future work in the conclusion.

Author Response

This manuscript produces Zn-Fe nanostructures with different morphologies: i) nano-alloys; ii) nano-layers; and iii) nano-layers combined with nanoparticles, aiming to produce materials with multifunctional properties. The idea is good and sounded; however, many issues should be addressed according to the following comments:

The overall presentation, readability, and more results and analysis are mandatory. Please, correct the language problems, it is weak from the Grammarly and sequences of events, I catch 22 errors by using a personal program, and the authors should cure them carefully.

Response: We thank the reviewer for your comments, which help us to improve the quality of the manuscript. The overall presentation and readability were revised, and more discussion and analysis were added along with the manuscript. The grammatical errors were also corrected using specialized software.

The "Abstract" section should be more intensively focused on the main idea directly and must contain the contribution of this manuscript supported with numerical result indicators. Also, please write the abbreviations of all nanoparticles " Zn and Fe" in the abstract section.

Response: The abstract has been improved. The main idea of the paper has been carefully highlighted and numerical indicators were included. Besides, all the abbreviations for the nanoparticles were included.

The "Introduction" section should be made much more impressive by highlighting your contributions. The novelty of this manuscript must be explained simply and clearly in points at the end of the introduction section. Note that, the introduction section should consist of three parts, i.e., a general introduction to the topic, followed by a literature survey, then the contribution clarifications.

Response: The introduction section was restructured and modified as suggested by the reviewer in order to highlight the novelty and the main contributions of our manuscript.

The introduction section should be enriched with up-to-date references by adding and citing the latest trends in the area of the influence of the nanoparticle's functionalization process on the morphology, structure, and characterization within the food packaging materials matrix. E.g., Effect of functionalized TiO2 nanoparticles on dielectric properties of PVC nanocomposites & Recent Advances in Polymer Nanocomposites Based on Polyethylene and Polyvinylchloride for Power Cables & PVC Nanocomposites for Cable Insulation with Enhanced Dielectric Properties, PD Resistance & Development of Industrial Scale PVC Nanocomposites with Comprehensive Enhancement in Dielectric Properties.

Response: More references were added to the introduction section.

It is mandatory to check carefully all the abbreviation definitions, symbols, and standard units in the whole manuscript. I catch some errors and the other symbols are not defined, please, define the abbreviations. Also, there are many standard tests without citing their references.

Response: All the abbreviations, symbols, and standard units were defined in the manuscript. References to the standard tests were also added.

The resolution and quality of figures must be modified; they should be presented as close to the camera-ready format (e.g., Figs. 2, 5, and 7). Also, please don't use the symbol abbreviations on X-Y-axes, they must have the full name with their SI units.

Response:  The resolution of the images was corrected. The symbols abbreviations in Figure 7 were corrected.

It will be helpful to the readers if a new subsection 3.3 is added that contains various discussions about the obtained results and other comparisons with previous studies.

Response: We appreciate the reviewer proposal. However, we believe that a discussion section led to repetition of the finding and a more complex structure of the paper, where the readers need to go back and forward in the paper. However, we have reviewed the document very carefully and more discussion of the results were included.

The conclusion section should be more concentrated and supported by the numerical results. Also, the authors may propose some interesting problems as future work in the conclusion.

Response: The conclusion section was improved as suggested by the reviewer.

Reviewer 3 Report

The authors have prepared multifunctional Zn-Fe nanostructures, including nano-alloys, nano-layers, and nano-layers combined with nanoparticles. The authors demonstrated that pure Zn has presented significant antibacterial activity against S. aureus and E. coli bacteria, as well as chromatic effects after oxidation. Furthermore, Zn-Fe nanostructures with different atomic arrangements have only revealed antibacterial effect and against S. aureus.

Overall, this work can inspire more material design ideas for food packaging materials. Therefore, I would like to recommend this work to publish in Nanomaterials. Below are a few suggestions for the authors.

1. What is the green pseudocolor in Figure 3? The authors should describe in main text or caption of Figure 3.

2. For “3.1.2. Antimicrobial properties”, the figure number should be corrected from Figure 7 to Figure 8.

3. The authors have demonstrated that pure Zn has revealed the best antibacterial activity. So, why do the authors prepare different Zn-Fe nanostructures?

4. For the introduction “One of the examples is metallic nanoparticles, which have been explored as antibacterial agents, antioxidants, and catalyzers, among others”, more references could be cited to broaden the introduction.

https://doi.org/10.3390/ijms20122924

Author Response

The authors have prepared multifunctional Zn-Fe nanostructures, including nano-alloys, nano-layers, and nano-layers combined with nanoparticles. The authors demonstrated that pure Zn has presented significant antibacterial activity against S. aureus and E. coli bacteria, as well as chromatic effects after oxidation. Furthermore, Zn-Fe nanostructures with different atomic arrangements have only revealed antibacterial effect and against S. aureus.

Overall, this work can inspire more material design ideas for food packaging materials. Therefore, I would like to recommend this work to publish in Nanomaterials. Below are a few suggestions for the authors.

  1. What is the green pseudocolor in Figure 3? The authors should describe in main text or caption of Figure 3.

Response: The meaning of the colors found in Figure 3 was described in the Figure caption.

  1. For “3.1.2. Antimicrobial properties”, the figure number should be corrected from Figure 7 to Figure 8.

Response: The number of the Figure relate with the antimicrobial properties of the nanostructures was corrected along of the 3.1.2 section.

  1. The authors have demonstrated that pure Zn has revealed the best antibacterial activity. So, why do the authors prepare different Zn-Fe nanostructures?

Response: As the reviewer said, the pure Zn system revealed great antimicrobial potential against both, Gram-positive S. aureus bacterium and Gram-negative E. coli bacterium, when compared with the systems containing Fe, which also showed a positive effect against S. aureus, but negative against E. coli. However, we produced different Zn-Fe nanostructured systems because we wanted to evaluate not only the antibacterial activity of the coatings, but also, other active (oxygen absorption capacity) and intelligent (chromatic changes) properties that they could have. In our case, we proved that the presence of Fe in the Zn NLs+Fe NPs coating significantly influenced on the chromatic functionalities of the sample, decreasing its opacity from 100 to 10%, and presenting a notorious color change, related to the accelerated iron oxidation due to its galvanic corrosion potential when subjected to high humidity environment.

  1. For the introduction “One of the examples is metallic nanoparticles, which have been explored as antibacterial agents, antioxidants, and catalyzers, among others”, more references could be cited to broaden the introduction.

https://doi.org/10.3390/ijms20122924

Response: The introduction has been modified accordingly and more references were added.

Round 2

Reviewer 1 Report

Accept in present form

Reviewer 2 Report

Most of my comments are adjusted